# Mobile Eye-Tracking Data Analysis Using Object Detection via YOLO v4

**DOI:** 10.3390/s21227668

**Published:** 2021-11-18

**Authors:** Niharika Kumari, Verena Ruf, Sergey Mukhametov, Albrecht Schmidt, Jochen Kuhn, Stefan Küchemann

**Affiliations:** 1Physics Education Research Group, Physics Department, TU Kaiserslautern, 67663 Kaiserslautern, Germany; kumari@rhrk.uni-kl.de (N.K.); vruf@physik.uni-kl.de (V.R.); mukhamet@physik.uni-kl.de (S.M.); kuhn@physik.uni-kl.de (J.K.); 2Mediainformatics Group, Institute of Informatics, LMU Munich, 80337 Munich, Germany; albrecht.schmidt@um.ifi.lmu.de

**Keywords:** eye movements, eye tracking, object detection, YOLO, Faster R-CNN, physics experiments

## Abstract

Remote eye tracking has become an important tool for the online analysis of learning processes. Mobile eye trackers can even extend the range of opportunities (in comparison to stationary eye trackers) to real settings, such as classrooms or experimental lab courses. However, the complex and sometimes manual analysis of mobile eye-tracking data often hinders the realization of extensive studies, as this is a very time-consuming process and usually not feasible for real-world situations in which participants move or manipulate objects. In this work, we explore the opportunities to use object recognition models to assign mobile eye-tracking data for real objects during an authentic students’ lab course. In a comparison of three different Convolutional Neural Networks (CNN), a Faster Region-Based-CNN, you only look once (YOLO) v3, and YOLO v4, we found that YOLO v4, together with an optical flow estimation, provides the fastest results with the highest accuracy for object detection in this setting. The automatic assignment of the gaze data to real objects simplifies the time-consuming analysis of mobile eye-tracking data and offers an opportunity for real-time system responses to the user’s gaze. Additionally, we identify and discuss several problems in using object detection for mobile eye-tracking data that need to be considered.

## 1. Introduction

Remote eye-tracking has been used in various forms in education research for the last 50 years and, at present, it is characterized by a high spatial (>0.15${}^{{}^{\circ}}$) and temporal resolution (<2000 Hz) [1,2]. Eye tracking offers the opportunity to analyze learning and problem-solving processes. The widespread use of stationary eye tracking is also attributed to its relatively straightforward data analysis, using commercial or open-source software packages [3,4]. More recently, mobile eye-tracking solutions became available, which offer the opportunity to extend the range of applications in comparison to those of stationary eye trackers to real field studies, such as lectures or classrooms conducting experimental work. Despite these compelling opportunities, and although mobile eye-tracking has been applied in some cases (see Reference [5] for a review), quantitative data analysis is often absent in such studies. Common approaches to quantitative data analysis include the use of markers close to the object and mapping fixations on a screenshot [6,7]. Mapping the fixations of several participants on a screenshot is very helpful to evaluate larger numbers of participants, but it is limited to specific situations in which the setting does not change during the experiment. This is due to the fact that manual labelling is a very time-consuming process and, therefore, not feasible for real-world situations in which participants move or manipulate objects and their view changes with almost every frame. Using markers close to the relevant object offers the advantage that fixations on objects can be quantified, but the assignment of an area of interest (AOI) to an object based on a marker is sensitive to the detection of the markers; i.e., if the markers are covered, the object would not be detected. Therefore, an automated method that identifies objects automatically for several participants would simplify the mobile eye-tracking data analysis of large datasets.

In this work, we address the use of object detection to assign students’ gaze data to real objects during an authentic experimental science laboratory course. For this purpose, we compare three object-detection models, namely You Only Look Once, version 4 (YOLO v4), YOLO v3, and a faster region convolutional neural network (Faster R-CNN). The trained models assign bounding boxes to the objects detected in each frame (see Figure 1). These bounding boxes are used as AOIs during the analysis of the eye-tracking data. All these algorithms allow for object detection in real-time that can be used to adapt the instructional material to the learners’ needs. In summary, we consider two aspects: (1) the machine-learning algorithm calculating bounding boxes for objects and (2) whether this algorithm could be successfully used to find the object participants focused on, which is the aim of eye tracking. Together with the several advantages, we specifically identify and discuss constraints that need to be considered when using object detection for the assignment of gaze data to objects. The code used here can be found at [8].

## 2. Relevant Literature

### 2.1. Object Detection

Object detection has been one of the core research areas in computer vision. Object detection models are used to automatically detect objects in images and videos (for an overview of the general approach, see [9]). While they are used in many contexts, such as facial pattern recognition [10,11] or object tracking [12], they have not been used to automatically detect objects when evaluating mobile eye-tracking scene videos during an authentic science laboratory course. In this study, we illustrate the use of object detection in mobile eye-tracking. Three models, YOLO v3 [13], YOLO v4 [14] and Faster R-CNN [15], are compared regarding their performance in detecting objects, their location, and the accuracy of gaze patterns as compared to a ground truth in the scene videos from mobile eye-tracking. You Only Look Once (YOLO) consists of multiple convolutional layers and two fully connected layers. The network converts the image into (S * S) grid cells at multiple scales and, for each cell, bounding boxes (B), together with a confidence score, are returned. Each bounding box is represented by four normalised coordinates (x, y, w, h) in the image and a confidence score. The confidence score reflects how confident the model is of an object being in the predicted bounding box, and how precise the prediction location is compared to the ground truth bounding box. A set of class probabilities *C* are also returned per grid cell. C refers to the conditional probability of each class given an object in a cell. Thus, the last obtained dimension is S * S * B * 5 + C [13]. Although YOLO v3 is the fastest detection framework, it suffers from low accuracy. However, the accuracy is well handled by the newer version, YOLO v4. YOLO v3 and v4 have been used previously in various applications [16,17,18,19,20,21].

Another frequently applied object detection model that uses regional proposal layers is a Faster R-CNN. Its prior versions, R-CNN and Fast R-CNN, used a selective search to predict bounding boxes. Selective search refers to the estimation of a region of interest (ROI). ROIs are regions in the image where an object could be located. Usually, thousands of ROIs are generated per image in order to increase the probaility of a positive prediction. These ROIs are then fed to the convolutional neural network for feature extraction. Using those features, a classification is performed. Fast R-CNN included ROI pooling, which transforms all the ROIs of different sizes to a fixed size, thus reducing the computaion. However, due to its multiple steps and selective search, R-CNN frameworks were slow to optimize. Hence, they are known to be accurate but are not suitable for real-time applications. Faster R-CNN introduced a Region Proposal Network (RPN), making it much faster than its previous versions. These models are known to be fast and exhibit a good mean average precision (MAP) score in object detection [13,22]. For an overview and comparison of object detection methods using deep CNN, see the review by [23]. We compare the performance of Faster R-CNN, YOLO v3, and YOLO v4 on the scene videos obtained from an exemplary experimental setting.

### 2.2. Mobile Eye-Tracking in Education Research

Mobile eye-tracking offers the opportunity to monitor learning processes in real environments. In this way, the technique allows for the investigation of social influences on learning [24]. It also enables study of the underlying processes that occur when students generate external representations [25]. For example, Hellenbrand and colleagues analyzed the learning strategies of students who generated schematic representations of virus cells and discovered that relevant words in the instructional text were reread more often, and transitions between the drawing and the text increased when students produced the drawings themselves [25]. Schindler and colleagues used mobile eye-tracking to investigate creative problem-solving processes in the context of math and found that this method allows for more insight than analysis of videos and/or paper-pencil solutions [26]. Mobile eye-tracking has also been used to analyze teaching activities while teachers were wearing head-mounted eye-trackers [27] that allow for the quantification of eye contact between teachers and students during scaffolding support [28]. Moreover, Jarodzka, Gruber, and Holmqvist pointed out that remote eye-tracking studies are usually performed in laboratory environments, where the participants are tested under optimal conditions [24]. This setting, however, neglects the influence of the presence of others, which might inhibit or facilitate performance and, therefore, demand more studies in real environments. One of the problems the authors mentioned is that the cumbersome analysis of dynamic real-world stimuli hinders the realisation of such studies [24]. In this context Maltese, Balliet, and Riggs used mobile eye-tracking during a geology field course, but found it “technically demanding and operationally difficult” to analyze the eye-tracking data ([29], p. 81). Eventually, the authors mostly used scene videos for the analysis and did not require eye-tracking measures [29]. A simplified analysis of mobile eye-tracking could also facilitate studies on visual attention during a lecture with a larger sample size [30], during lecture demonstration experiments [31], or during students’ experiments [32].

### 2.3. Object Detection in Mobile Eye Tracking

The usefulness of object recognition in mobile eye-tracking has been previously recognised, as it can facilitate and enhance the analysis of visual behaviour in the real world [33]. There has been research on the detection fixations in a 3D environment using fiducial markers, using geometric modelling and overlaying real-world objects with augmented-reality models (EyeSee3D method) [34]. In a previous study, De Beugher, Brône, and Godemé compared the SIFT, ASIFT, and SURF algorithms regarding their suitability for object detection in mobile eye-tracking data and achieved promising results regarding various performance measures, such as efficiency and speed [35]. Furthermore, de Beugher and colleagues presented a method to analyze mobile eye-tracking data via Oriented Fast and Rotated BRIEF (ORB) and a novel technique for temporal smoothing to reduce false-positive and false-negative detections [36]. This method did not require a separate training and could detect occluded people (only torso/upper body visible) via a Deformable Parts Model [36]. Based on this, they described a way of detecting faces and people, as well as objects, using computer vision techniques [37]. In this case, a Deformable Part Model and a Haar-based model, along with a Kalman filter, were used to detect faces and people; ORB features were also employed for object recognition [37]. Researchers have also developed methods to analyse mobile eye-tracking data. For example, Essig and collegues developed the JVideoGazer that automatically analyzes and annotates video data [38]. They used translation, as well as scale and rotation invariant object detection (SURF), in combination with a Fast-Hessian detector and Principal Component Analysis [38]. Kurzhals, Hlawatsch, Seeger and Weisskopf developed an “interactive labeling and analysis system” ([39], p. 301) to analyze and annotate mobile eye-tracking data [39]. They extracted a thumbnail image based on the gaze coordinates, then carried out video segmentation, and finally compared and clustered images using SIFT features with a bag-of-features approach and employed self-tuning spectral clustering based on a maximum number of AOIs [39]. In this way, scarf plots and attention histograms could be computed as output [39]. The current research also uses eye-tracking in virtual reality [40]. This should, in theory, be adaptable to real-world environments by using artificial reality and overlaying real objects with virtual colliders. It is also possible to replicate a real environment in a virtual setting. However, both methods require access to the respective hardware and have some limitations, such as discomfort when moving in virtual environments [40].

It would, therefore, be simpler if already available hardware and easily adaptable software could be used to analyze mobile eye-tracking data. With the development of new object detection algorithms, such as YOLO, the results of previous research might be improved. All of the previous methods did not allow for object detection in real-time without delays. Consequently, they do not offer the opportunity for the real-time adaption of digital learning material, scaffolding support, or real-time feedback for teachers. These works also do not include an in-depth discussion of the problems of assigning gaze data to objects via object detection. Furthermore, these works do not combine object detection in a real educational laboratory setting, i.e., it is unclear how they perform in authentic/real experimental scenarios. This work aims to fill these gaps.

## 3. Methods and Materials

### 3.1. Participants and Procedure

The study was conducted as part of an experimental lab course on the functioning of the human eye in a health science study program. The study consisted of two phases, in which N = 42 first-year health-science students learned how the image of a converging lens is created by constructing optical beam paths.

In the experimental lab course, the students were asked to solve four experimental tasks. For this purpose, they received material with an exemplary illustration of the schematic optical path, a real image of the setup including the labels and the experimental component. In both phases, students’ gaze data was recorded using mobile eye-tracking.

The experimental setup, including some of the components that needed to be detected, is shown in Figure 2a. Figure 2b,c present how students worked on the paper-based tasks and on the experimental setup, respectively, which both included elements that were intended to be detected.

### 3.2. Students’ Lab Course

Automatic object detection was used to assign fixations to the corresponding objects during a students’ lab course concerning optical lenses. Prior to the students’ experiment, participants were instructed about light distribution and how lenses disperse light beams, sometimes generating their own representation of this process. Both instruction and experiment concerned the process of light passing through a lens, the lens focusing the light and a picture appearing upside down on, for example, a paper. The experiment consisted of a power supply that was connected to a light source. From this, the light went through an arrow-shaped form. It was then focused and dispersed by a lens, and a resulting image could be seen on a piece of paper. All parts of the experiment, apart from the power supply, could be moved on a slider that was also movable.

### 3.3. Eye Tracking

We used a Tobii Pro Glasses 2 with 50 Hz and a scene camera of 1920 × 1080 pixels and 25 fps. We calibrated the Tobii Pro Glasses 2 for each participant via a 1-point calibration before the start of the experiment. The quality of the calibration was verified with 5 points distributed on an A3 paper. There was one dot in the middle, with another four dots at each corner. In this way, the outer dots were approximately 24 cm apart from the dot in the centre. The paper was placed on the table and participants looked at it from an approximately 50 cm distance. The viewing angle was, therefore, $\theta =25.6^{{}^{\circ}}$. If the calibration failed, meaning that participants’ eye movements did not coincide with these five points on the paper, the calibration was repeated until a successful calibration was achieved. At this point, no subject had to be excluded due to failed calibration. Figure 3a shows a sheet of paper with a photograph of the setup, including labels of the experimental components, a schematic of the beam paths and three experimental tasks. Figure 3b shows the identified bounding boxes on this sheet of paper in one frame. On the paper, we distinguished different tasks (right-hand side of the paper) as well as different types of instruction, such as text, picture, and graphical representation (left-hand side of the paper).

### 3.4. Processing of Eye-Tracking Data

Once the gaze data were collected by mobile wearable eye-tracking glasses, they could be overlaid on the first-person video footage to match them with the objects seen by the participant across the timeline of the experiment. In this step, an I-VT method was used to classify fixations. In the case of the Tobii Pro Glasses 2, the assignment of gaze data to parts of the text and images could be carried out manually or automatically in the case of static settings using the software Tobii Pro Lab [7] via mapping fixations on a reference image of the stimuli.

In the case of manual mapping, the assignment was made by a human going through all the fixations and indicating the gaze location on the reference image. The reference image is typically a photograph or symbolic depiction of the setup, on which fixations are mapped independently of possible manipulations of the setup or head movements that lead to a shifted field of view. When using automatic mapping, the comparison is made by computer vision algorithms, which have the task of finding reference image features in video frames, and then determining the fixation point within them. This automatic mapping in Tobii Pro Lab allows for the assignment of the corresponding gaze position and creation of gaze maps on static stimuli such as sheets of paper. At the same time, due to natural difficulties, such as blurred video frames due to fast movement, an incomplete/partial image of a snapshot in a video frame, or partial occlusion by a human hand, errors appear in almost every video-recording of an experiment, which have to be manually corrected.

Automatic mapping does not work for images of three-dimensional objects whose representation is highly dependent on the angle of view, or for changing objects in a dynamic environment. In the case of changing objects in a dynamic environment, such as the physics experiment, the manual mapping feature provided by Tobii Pro would also require that the reference image constantly changes, similar to a reference video, which, in turn, would be different for each participant. Alternatively, the fixations could be mapped on an abstract image containing representations of each object. In both cases, this would require going through each fixation and manually assigning it to the objects, which is very time-consuming but, at present, the only option [41,42]. This is also problematic, as we would like to quantify gaze shifts between (parts of) the instruction, individual tasks, and experimental components.

In comparison, object detection not only allows for the automatic identification of AOIs and assignment of the fixation, it also allows for the quantification of the relative position of objects. During a physics experiment, the latter feature is particularly useful in the identification and digitization of electrical circuits or the quantification of the haptic interaction with experimental components.

### 3.5. YOLO v4

YOLO v4 is the upgraded version of YOLO v3 and is significantly faster and more accurate. Various additions and modifications have been made to the network that contribute to its efficiency. YOLO v4 consists of CSPDarknet53 [43] as a backbone. CSPDarknet53 is a feature-extraction network, which splits the input into two parts. One part goes through DenseNet and the other part does not go through any convolutions. Due to its increased receptive fields, Spatial Pyramid Matching (SPP) [44] and Path Aggregation Network (PAN) [45] are used as Neck. The SPP layer performs maxpooling to generate fixed size output representations, and is used after the feature extraction to avoid the limitation of a fixed size input requirement. The role of the neck is to aggregate features from different stages. The SPP block is added over the CSPDarknet53, since it separates the most significant context features and causes almost no reduction in the network operation speed [14]. Due to the high abstraction of information in the higher-level features, it is difficult to make predictions for each pixel. PANet helps the information flow by performing pooling at different levels, thus preserving spatial information [14]. YOLO v3 is used in the third stage as the head, where the usual object detection and localization occurs. A new data augmentation method is used, called mosaic, which mixes four different training samples. This reduces the need for large mini-batches and provides better object features. Another augmentation method used is Self Adversarial Training (SAT). SAT alters the training image and trains the network to identify objects in the altered image. Besides these new additions to the network, several other modifications were implemented to improve the efficiency of the model [14]. An overview of the YOLO v4 architecture can be seen in Figure 4.

### 3.6. Faster R-CNN

Faster R-CNN is based on Convolutional Neural Networks (CNN), relying on a Region Proposal Network (RPN) for region detection within an image [15]. RPN consists of a regressor and a classifier. For each pixel in the feature map, the RPN returns the objectness score of a target object in that location. The regressor predicts the bounding box coordinates. It uses multiple scales; hence, its predictions are translation-invariant (see Figure 5). The CNN is then used to refine and classify the proposed boxes [15]. The loss is calculated for both the RPN and the CNN via the weighted sum of locations and classification losses [15]. The runtime of Faster R-CNN is better than its previous versions. However, it is still expensive compared to both YOLO v3 and YOLO v4.

### 3.7. Bounding-Box Interpolation

Object detection in sequential frames is a tracking problem. The bounding boxes of a class might not be strictly detected in a continuous range of frames. To fill these gaps, interpolation is required. Interpolation is the process of estimating the unknown predictions (in our case, bounding boxes) from the known predictions. This process reduces the false negatives. For instance, in frame *t*, object A has been detected, but in frame t+1, it was not detected. Using interpolation methods, we can estimate the shift in coordinates from frame *t* to frame t+1. Once the shift is known, we can generate the missing bounding boxes in frame t+1. The interpolation of bounding boxes was performed by estimating the optical flow between two consecutive frames using the OpenCV Lucas-Kanade optical flow algorithm [46]. Lucas-Kanade optical flow is widely used for the robust estimation of optical flow between two images. The optical flow refers to the shift in the pixels of an image from time *t* to t+Δt. Considering a pixel *p* centred at the location (u,v) and time *t* in a 2D image. In any consecutive frames, the pixel *p* could be anywhere, due to the shift in the scene. The optical flow algorithm estimates the location of the pixel *p* in any consecutive frame at time (t+Δt).

The Lucas-Kanade method has previously been used for eye-tracking [47]. In our method, we also used this feature-tracking method to track similar features over consecutive frames for a bounding-box interpolation (see Figure 6). The entire set of video frames was divided into chunks, where a single chunk comprised five consecutive frames. The interpolation was performed within every chunk, and each chunk was independent of the other chunks. We chose five frames, based on the assumption that a fixation on one spot lasts at least 200 ms, which corresponds to five consecutive frames considering a frame rate of 25 Hz. We used this argument for the fixation duration because it is an indicator of how rapidly the environment changes, even though a fixation itself is not related to the interpolation of bounding boxes. We also varied this number of five frames within one chunk and verified that five frames led to the best performance of the algorithm. Chunks comprised of blurred frames were ignored during the interpolation. The blurred frames were identified using the variance in the Laplacian method. We also used a threshold of five frames to catergorize blurred images. In this way, the gaze was assigned to all frames, including the false negatives (see Figure 6). If there were five or more blurred images in a row, we discarded the bounding box. Models such as Deep Sort were not used, since they are used for the tracking of single objects and are more suitable for radar data. Using such models could lead to a high number of false-positive identifications.

### 3.8. Training the Model

The training sample was generated by taking additional images with another camera of the experimental setup and the individual objects in different perspectives and at various distances. To increase the robustness of the model, image augmentation techniques such as rotation, flipping, Gaussian blur, cropping, noise injection and background clutter, were used to create a more generalised training dataset for better feature extraction. The manual annotations of the 1000 training images were made using the labelImg software [48]. Taking the photographs, picture augmentation, and labelling took, overall, around 40 h of work. The number of classes was the number of objects in the experimental setup, e.g., the arrow, the graph, the lens, the light source, the paper, the picture, the power supply, the questions, the rail, the text, and others. The resolution of the training images was 416 × 416 px. 70% of the images were for training and the rest 30% were used for testing. New anchors were generated using the k-means clustering algorithm, and 26 anchors were used in total. The anchors were the predetermined initial bounding boxes of the objects that contained the information on the aspect ratio (the ratio of height and width) and were provided to the network. Hence, instead of generating new bounding boxes, the network only rescaled the size of the nearest anchor to the size of the objects. This feature makes YOLO v3 faster than other object-detection networks, which mostly use the Region Proposal Network (RPN) to detect the objects. The model was trained up to 34,000 iterations. The weights of the top 137 layers in the YOLO v4 pre-trained model were used to train the YOLO v4 model. The pre-trained model [49] was previously trained on the MS COCO dataset [50] and had a good MAP score of 75%. For the Faster R-CNN, we used the inception v2 model [51], which was pretrained on the MS COCO dataset. This was assumed to reduce the training time, and also limited the number of training images needed. However, it could also lead to a slightly worse performance than algorithms that were trained solely on a dataset developed specifically for this experiment. As our aim was to find a method that could be adapted to many mobile eye-tracking contexts, we decided to accept this limitation in favor of a faster training process and increased replicability.

### 3.9. Testing the Performance

The final trained model was tested on 42 videos on google colab, which allocates a varying amount of GPU, CPU, and memory to certain processes and supports CUDA 10.0. Therefore, we cannot state the exact processor required to run the model. All videos were MPEG-4 videos with a resolution of 1080 × 1920 px. An exemplary video V1 (Tobii Pro Glasses 2) frame can be seen in Figure 2b. The bounding boxes for the respective classes were generated (see Figure 1) and the gaze points for each video were extracted using Tobii Pro Lab [7]. The video of one exemplary participant was chosen for testing. The video consisted of 8222 frames. A total of 5953 frames were manually annotated with bounding boxes using the Darklabel software; this took 68 h. During annotation, a rectangular frame tightly fitted to the edges of the object was manually identified. All objects that were covered by another object or a human hand were only annotated if the main part of the object was visible. Manually annotating the location of the gaze points in a single video recording of approximately 7 min 19 s, consisting of 10,978 frames, took 12 h. This means that labeling all 42 participants would take about 500 h. Taking the time and effort required annotation and limited resources into account, we decided that this amount of data was enough for a reasonably accurate judgement of the algorithm’s quality.

## 4. Results

The objects were detected using the trained models. The gaze coordinates that were originally extracted from Tobii Pro Lab were used for estimating where the participant was looking at a certain timestamp. If the gaze coordinates were within the bounding box of an object A in the *i*th frame, we concluded that the participant was focusing on object A in the *i*th frame. In this way, we could determine the fixation (location where a person was focusing on) in each frame and, as a result, obtain a time series of all participants’ fixations. This time series data could, for example, be used for analysing the study patterns. The predicted fixation was compared with the ground truth fixation. The ground truth was obtained by manually looking at each video frame and recording the objects on which the gaze was focused at each timestamp. For the YOLO v4 model, the macro-precision score was 0.89 and the recall was 0.83. For the Faster R-CNN, the macro-precision score was 0.83 and the recall was 0.73. The confusion matrix for YOLO v4 with interpolation is displayed in Figure 7. In the case of YOLO v4, the true gaze was on the rail but predicted as "Other" 79 times, and it was predicted as paper 5 times. The interpolation of the boxes led to a significant increase in accuracy. For YOLO v4, we show the results without the interpolation step in Table 1 and the confusion matrix in Figure 8. There was an increase in the weighted F1 score, from 0.58 to 0.87, after interpolation. The confusion matrix of the Faster R-CNN can be viewed in Figure 9. For the Faster R-CNN, the gaze was on the rail but was predicted as “Other” 125 times and predicted as paper 62 times. This might be due to the closeness of these objects in the experimental setup (see Figure 2). The results for all objects for YOLO v4 and Faster R-CNN can be seen in Table 2 and Table 3; micro averages calculate the average of the contribution of all classes, which is useful for imbalanced classes and corresponds to the algorithm’s accuracy, whereas macro averages estimate the average across class averages, independently of their contribution, and weighted averages weigh the average by the number of samples in each class. Since the bounding boxes did not have irregular shapes, the overlapping of bounding boxes could not be avoided. We also tried to change the angle of the rectangular bounding boxes based on the angle change in the scene. However, this did not lead to any significant increase in the accuracy.

Despite the problem of overlapping boxes, an F1 score of 0.85 (Table 2) was obtained, which shows the advantage of using object detection in experiments automatically assigning students’ gaze patterns to real objects. YOLO v4 not only provided good scores, it was also comparatively faster than Faster R-CNN and YOLO v3, with 25 frames per second for a 500 MB video, as compared to 11 fps in case of YOLO v3, and a Faster R-CNN (see Table 4). This shows that YOLO v4 could easily be used in real-time scenarios, leading to fast and accurate results.

## 5. Discussion and Outlook

Typically, students’ lab courses are standard elements in STEM curricula and, therefore, play an important role in education. The understanding of the underlying experimental processes of students and strategies to solve the experimental tasks can be analyzed via mobile eye-tracking. Due to the dynamic setting and moving objects in the experimental lab course, standard automated mobile eye-tracking analysis tools are not practicable. Therefore, in this work, we compared three recently developed object detection algorithms, namely, YOLO v4, YOLO v3 and a Faster R-CNN, to assign students’ gaze to experimental components during a physics lab course. To our knowledge, this work reports the only ecologically valid setting in a science lab course, in which we account for the natural behavior of students, including the unexpected movement of objects, objects that are covered by other objects or body parts, and non-ideal camera perspectives. This allows us to draw conclusions on the suitability of object detection when assigning gaze data to objects in an authentic educational setting.

Regarding the work-time investment, taking the photographs, augmenting and labelling them took about 40 h, which is about one twelfth of the time needed in the alternative procedure of manually mapping the gaze points to the objects, which requires an estimated time of 500 h. Even if we account for the additional time needed to train and apply the algorithm (depending on the computer’s processing capacity), the total time needed to apply the object detection algorithm is considerably shorter than the manual alternative. Here, it is important to consider that the time needed for the manual mapping of gaze points linearly depends on the number of participants, i.e., the time of recorded eye-tracking data, whereas the time needed to use the object detection algorithm mainly depends on the number of objects to be detected. This means that using the object detection algorithm becomes more efficient with an increasing number of participants and a decreasing number of objects. Naturally, it would not be efficient if there is a large number of objects and small number of participants (time of recorded eye-tracking data).

Overall, we found that the YOLO v4 algorithm provides a better detection of all components throughout the experimental setting than the other two algorithms, but the main advantages were noticed in the detection of the rail and the questions.

We found the highest precision, recall and F1-scores for the YOLO v4 algorithm in comparison to the Faster R-CNN and the YOLO v3 model. The architectural differences mean that YOLO v4 is a faster and more robust model.

YOLO v4 reaches an F1 score of 0.85, which means that, considering all predictions, the sum of false-negative and false-positive predictions is about 17.5% of the true-positive predictions. To obtain a better idea of what this means, it is necessary to look at the false-positive and false-negative predictions for each label. For example, YOLO v4 reaches an F1 score of 0.69 for the light, which is the consequence of a high number of false-negative predictions of this object (a recall of 0.57). Similarly, the detection of the lens and the detection of the picture were also affected by an enhanced number of false-negative predictions (recall below 0.8). This means that the object was not detected when it was in the field of view of the participant. The reason for this could be that the light and the lens were both part of the setup and, in our case, the parts of the setup were frequently covered by each other. This explanation is also confirmed by the fact that, for both objects, YOLO v4 reaches a high precision, above 0.8, which means that the number of false-positive predictions is rather small, i.e., the objects were rarely confused, even though they look similar. This may be a consequence of the colorful markers we attached to the objects, making it easier to discriminate between them. From this observation, we can conclude that the partial coverage of objects is one of the main aspects that affects the F1 score of YOLO v4. This problem also becomes importan when evaluating the eye-tracking data, as it results in overlapping bounding boxes (see discussion below).

In general, using object detection to assign gaze points to objects causes a number of constraints that need to be considered independently of the algorithm:Variety of shapes: The shape of real objects is not always rectangular, but the bounding boxes are. This implies that fixations close to the edges, but not on the object, may be counted as fixations on the object. In other words, the spatial resolution of eye tracking is reduced.Perspective: Objects might be viewed under a certain angle, which may translate to a shape change when projected on a 2D plane, as in a video. This can result in the bounding box being either larger or smaller than the object. In the first case, the effect would be the same as the “variety of shapes” error, where the spatial resolution of eye tracking is reduced. In the latter case, fixations on the edges of objects might be missed.Overlapping bounding boxes: A disadvantage of using rectangular bounding boxes during object detection is that they may overlap (see Figure 3). This may lead to the fixation being assigned to more than one bounding box. Possible reasons for this are those mentioned above or when objects are partially behind each other, but still detected by the object-detection algorithm. This causes a conflict for the algorithm, because the eye can only focus on one object at a time. One solution to this problem would be to calculate the distance of the gaze from the center of each of the overlapping boxes, and the gaze would be allocated to that object closest to the center. However, this solution would lead to other difficulties, and it is part of a future work to identify which type of bounding boxes work best during mobile eye-tracking in experimental lab courses. Furthermore, while calculating the accuracies, we penalized our scores in the case of multiple gaze assignments. For instance, if the model predicted that the gaze was on the rail, the paper, and the lens within one frame, but in the ground truth the gaze was only on the arrow, a false-positive would be added thrice rather than once. If the gaze in the ground truth was on the rail and the model predicted the gaze to be on the rail, the arrow, and the lens, we would count one hit and two false-positives. In other words, if the gaze falls in a region where multiple bounding boxes overlap, we count one false-positive for each object to which the gaze is falsely assigned.False negatives: When objects in the users’ field of view are undetected, it effectively causes a reduction in the gaze detection rate according to the rate of correctly identified objects in the specific video. Therefore, we recommend setting a stricter threshold, compared to the currently used threshold to classify fixations, so participants’ gaze data is included in the analysis based on the gaze detection rate if there is no assisted object detection, as mentioned below.False positives: The object may be detected at places where it is not located. This may cause a gaze point to be assigned to another object.

These problems may cause errors that are either statistical or systematic in nature. If they cause statistical errors, they may enhance or reduce any quantitative comparison between participant groups, i.e., the variation in the number of occurrences of each error between participants is expected to follow a Gaussian distribution. When using object detection to assign gaze points to objects, this implies an additional error when making quantitative comparisons, which might affect significance testing. Therefore, we suggest reducing the significance threshold of the *p*-value in small sample sizes, in which the tested eye-tracking metric is not normally distributed. The reason for this is that, in normally distributed samples, the determination of the p-value (for example, via a *t*-test) considers a change in the sample’s variance. In not-normally distributed samples, there is an increased chance of coincidental and significant test results. Of course, this reduction in the significance threshold needs to be added to a correction for multiple testing, such as the Bonferroni method [52].

To reduce the constraints mentioned above, we recommend implementing a tracking feature that reports the objects that were tracked in each frame. This information can then be used to manually check the performance of the object detection and perform an assisted manual object detection, such as the assisted mapping tool implemented in Tobii Pro Lab [7].

Furthermore, for the first three errors, object segmentation algorithm may help to reduce these issues; however, this may provide a heavy computation time [53]. It might be possible to optimise this via other deep learning models [23].

Additionally, the abovementioned problems may cause systematic errors, for instance, in the following two specific settings. First, if relevant objects regularly cover each other during the recording, the problem of overlapping bounding boxes would become a systematic error. This means that, when making quantitative comparisons of gaze data in such settings, object detection is prone to errors, and we do not recommend using it. Second, in settings in which objects are very similar to each other, there may be an increased number of false-positive assignments. In such cases, we recommend implementing salient markers as distinctions between the objects to ensure a reliable object detection.

Regarding the detection rate, the YOLO v4 algorithm reaches the highest speed of 25 fps, which opens opportunities for real-time object detection. In STEM education, this advantage provides the potential for the real-time determination of students’ competencies and the subsequent adaptation of learning material based on students’ gaze data.

In conclusion, the YOLO v4 algorithm reaches the highest accuracy during object detection from a scene camera of a mobile eye-tracker. This allows for the robust assignment of gaze data to real objects in the identification of the relative position of objects in real-time, which provides the potential for adaptive learning systems that operate in a real environment based on students’ gaze data. Based on the results, we recommend using YOLOv4 for the automatic assignment of eye-tracking data to objects if (i) the settings are evaluated regarding the systematic errors mentioned above, and (ii) the total recording time of all participants is long enough (due to the large number of participants) in comparison with the number of objects used, so that the time necessary to implement the algorithm is clearly shorter than the estimated manual mapping time.

As eye-tracking is also frequently implemented in new and emerging technologies (such as AR-smartglasses, see [40,54,55]), this work opens opportunities for an efficient analyses of complex eye-tracking data in these technologies.

## Figures and Tables

**Figure 1 sensors-21-07668-f001:**
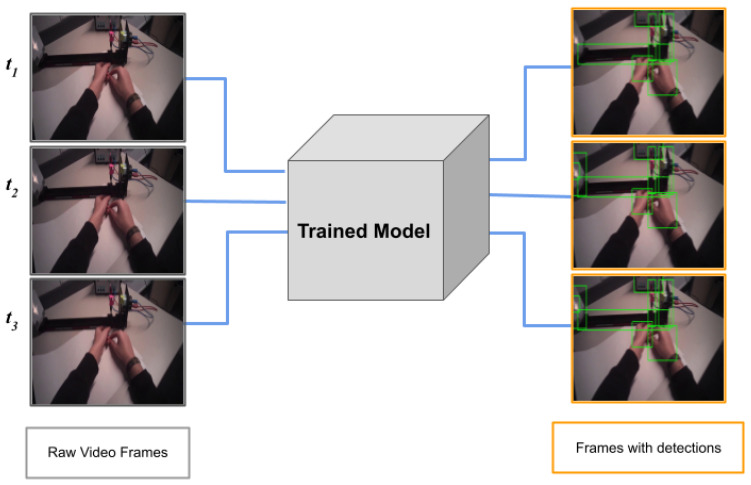
Each frame of the video is passed through the trained model (for more details, see Section 3.5), resulting in multiple predictions (bounding boxes) at each timestamp. The figure above shows object detection for three consecutive frames of a video.

**Figure 2 sensors-21-07668-f002:**
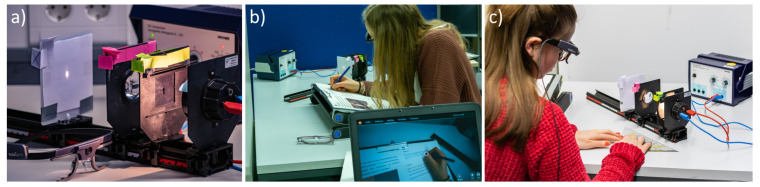
(**a**) Experimental setup; (**b**) An example of the study setup for participants with the experiment in the background, a participant working on the instructional material, and the computer monitor of the supervisor in the foreground; (**c**) a participant working with the experiment.

**Figure 3 sensors-21-07668-f003:**
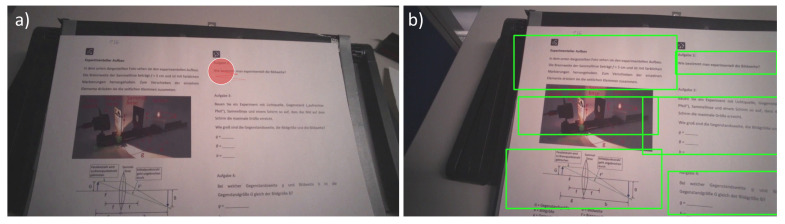
(**a**) A frame from a recorded experimental video with gaze focused on questions. During training and testing, the gaze is excluded from the video. (**b**) A frame from an output video with bounding boxes detected by the trained model.

**Figure 4 sensors-21-07668-f004:**
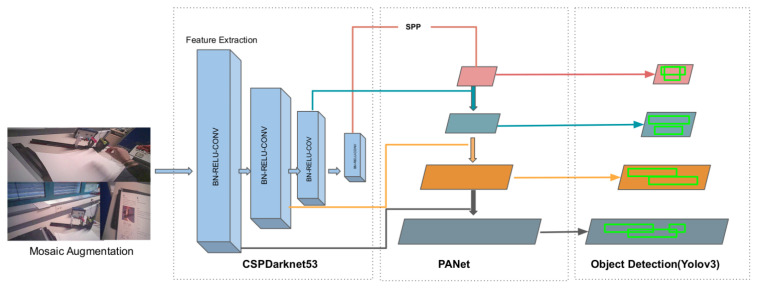
Architecture of YOLO v4.

**Figure 5 sensors-21-07668-f005:**
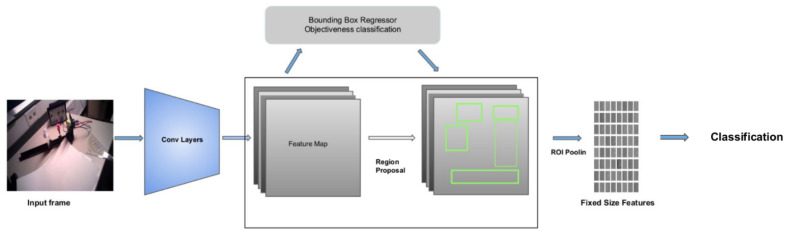
Architecture of Faster R-CNN. Figure adapted from [15].

**Figure 6 sensors-21-07668-f006:**
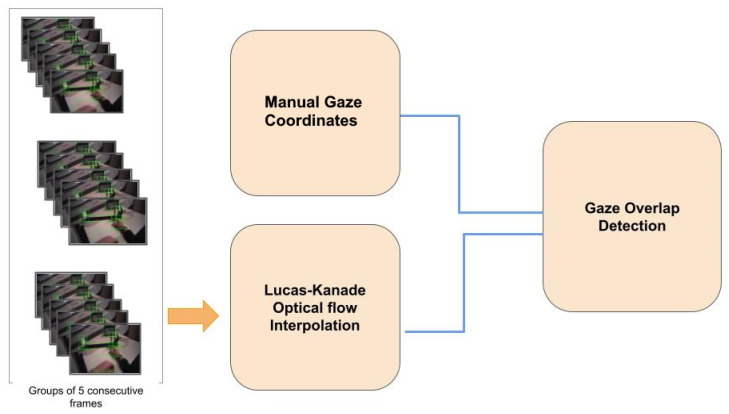
Schematic of the assignment of the manual fixation coordinates to the bounding boxes for five consecutive frames.

**Figure 7 sensors-21-07668-f007:**
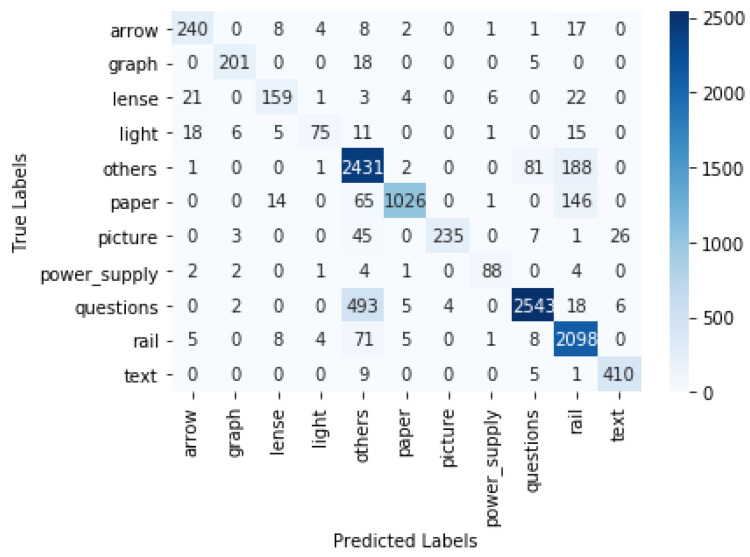
Confusion matrix for YOLO v4 with interpolation.

**Figure 8 sensors-21-07668-f008:**
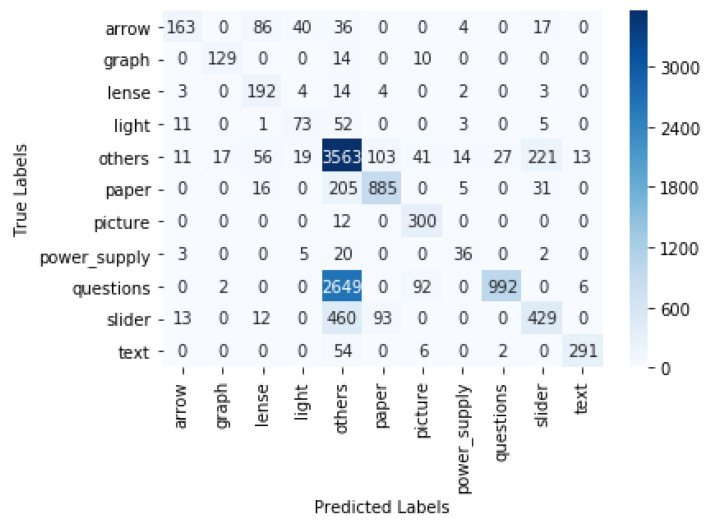
Confusion matrix for YOLO v4 without interpolation.

**Figure 9 sensors-21-07668-f009:**
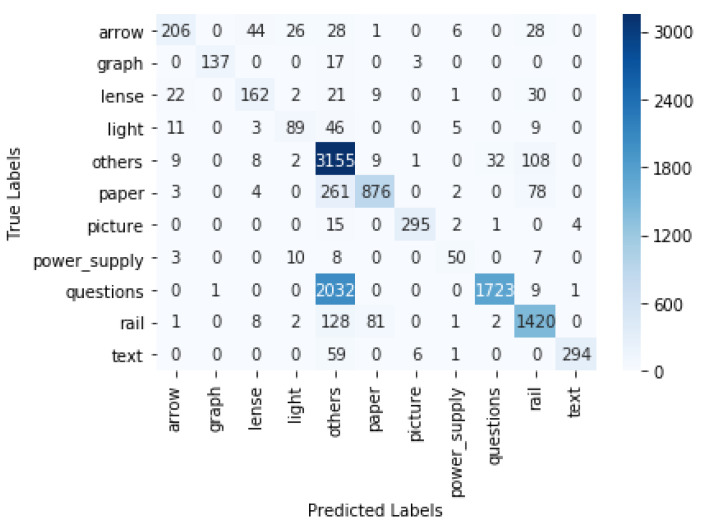
Confusion matrix for Faster R-CNN with interpolation.

**Table 1 sensors-21-07668-t001:** Confusion Report for YOLO v4 without interpolation.

	Precision	Recall	F1 Score
Arrow	0.80	0.47	0.59
Graph	0.87	0.84	0.86
Lens	0.53	0.86	0.66
Light	0.52	0.50	0.51
Others	0.50	0.87	0.64
Paper	0.82	0.77	0.79
Picture	0.67	0.96	0.79
Power Supply	0.56	0.55	0.55
Questions	0.97	0.27	0.42
Rail	0.61	0.43	0.50
Text	0.93	0.82	0.88
Micro avg.	0.61	0.61	0.61
Macro avg.	0.71	0.67	0.65
Weighted avg.	0.73	0.61	0.58

**Table 2 sensors-21-07668-t002:** Confusion Report for YOLO v4 with interpolation.

	Precision	Recall	F1 Score
Arrow	0.84	0.85	0.85
Graph	0.94	0.90	0.92
Lens	0.82	0.74	0.78
Light	0.87	0.57	0.69
Others	0.77	0.90	0.83
Paper	0.98	0.82	0.89
Picture	0.98	0.74	0.85
Power Supply	0.90	0.86	0.88
Questions	0.96	0.83	0.89
Rail	0.84	0.95	0.89
Text	0.93	0.96	0.95
Micro avg	0.87	0.87	0.87
Macro avg	0.89	0.83	0.85
Weighted avg	0.88	0.87	0.87

**Table 3 sensors-21-07668-t003:** Confusion Report for Faster R-CNN with interpolation.

	Precision	Recall	F1 Score
Arrow	0.81	0.61	0.69
Graph	0.99	0.87	0.93
Lens	0.71	0.66	0.68
Light	0.68	0.55	0.61
Others	0.55	0.95	0.69
Paper	0.90	0.72	0.80
Picture	0.97	0.93	0.95
Power Supply	0.74	0.64	0.68
Questions	0.98	0.46	0.62
Rail	0.84	0.86	0.85
Text	0.98	0.82	0.89
Micro avg.	0.72	0.72	0.72
Macro avg.	0.83	0.73	0.76
Weighted avg.	0.81	0.72	0.72

**Table 4 sensors-21-07668-t004:** Comparison of scores between three object detection algorithms.

	Precision	Recall	F1 Score	FPS
YOLO v4	0.89	0.83	0.85	25 fps
Faster R-CNN	0.83	0.73	0.76	11 fps
YOLO v3	0.77	0.64	0.61	10 fps

## Data Availability

Not applicable.

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
