# Peer review of "Mobile Eye-Tracking Data Analysis Using Object Detection via YOLO v4"

_sensors, 2021, doi:10.3390/s21227668_

Round 1

Reviewer 1 Report

The authors present a thorough revision, addressing all issues raised. Specifically, the discussion is largely improved and the section on future studies is insightful.

Author Response

Reviewer’s comment: The authors present a thorough revision, addressing all issues raised. Specifically, the discussion is largely improved and the section on future studies is insightful.

Authors’ response: Thank you for your assessment of our manuscript.

Reviewer 2 Report

The authors present a study of automatic gaze classification, comparing three object detection algorithms. While interesting and sorely needed in the field, the study as is leaves several significant things to be desired, which I set out below.

It is unclear to me what the outcome of the study is. Its aim seems to be a feasibility study (line 6 “we explore the opportunities”), but no conclusion is reached regarding whether performance of any of the three methods is good enough. What does an F1 score in the 0.80s mean? To allow this study to actually make a point, the authors need to describe it in terms of a scenario. A researcher has the goal to study X (real example research question that their data set allows to answer, can be a toy example, no need for ground breaking stuff of course, focus is on the method here), and the study can then check how any of the proposed methods weigh up to the current practice/golden standard (manual coding). Are the same conclusions reached? That allows the reader to see if any of the methods tested are usable in practice.

Open methods: will you provide at least the code used to run the study so that others can try and apply it in practice? Right now its little more than a “yay we succeeded” message, others can’t do much with it. Even better would be to provide the data as well so that the authors provide a complete working example, and the results in the paper can be replicated.

Unclear methods:

  1. You make claims about performance but do not at all describe the hardware used?
  2. You have overlapping bounding boxes and thus gaze that may be flagged as falling on multiple objects at the same time. How do you deal with this problem?
  3. You talk about fixations (e.g. line 154), gaze locations and gaze focuses (whats that?): what is it that you actually use for the study? Individual gaze samples or classified fixations? If the latter, how did you classify? If the former, say that explicitly. Use one word throughout.
  4. It seems that you both manually annotated bounding boxes on video frames, and labeled gaze data with objects looked at (always only one? How did you decide if unclear? Did gaze always have to be within object edges? Make this way more clear). What were the manual bounding boxes and gaze labels used for? Why manually label gaze if you already have manual object bounding boxes that can be used to do that labeling?
  5. Should performance of the algorithm be scored based just on the bounding boxes it produced, compared to your manual bounding boxes? Evaluation using gaze seems potentially problematic, especially given that you use only one participant, as a method is not penalized for a missed object or wrongly placed bounding box if that object is not looked at.
  6. What training/testing/validation splits were used?
  7. What did the interpolation actually do? Fill gaps where an object is not detected on an intermediate frame? Smooth out corner positions? Not said at all…

Vague writing, unclear logic, typos and word choice, and minor things:

2: “extend the range” as compared to what?

21: “use of eye-tracking studies” is a weird phrase, skip “studies”? who does the attributing referred to in that sentence? I have not heard that claim in the context of mobile eye tracking, and your references there do not support it.

33: “but such an assignment of an area of interest (AOI) to an object based on a marker needs to be done for each participant separately which also makes it a complex task” instead of complex you mean time-consuming? Also, why would it have to be done for each participant separately? I presume you would use a constant marker-object pairing throughout the experiment, and thus a simple computer script can do this work for you?

40: “The trained YOLO v4 model assigns bounding boxes to the detected objects in each frame” what do the others do?

Fig 1 is of insufficient quality, I can’t see almost anything

64: “less accurate”: than what?

64: “this problem”: what problem?

117: “this kind of data”: what kind of data?

154: “The automatic object detection was used to compute fixations for conceptually relevant objects” remove that part of the sentence, it doesn’t belong there. Presumably this paragraph is to describe the recording setting/procedure.

155: “before the experiment”: you’re not doing an experiment, what are you referring to? This term appears in many places in the manuscript, it is confusing.

164: “that”: what? The scene camera? I assume each participant?

167: report the observed calibration accuracy. Also describe this paper better. How eccentric (in viewing angle) were the five points, from what distance were they viewed?

176: “it would not allow”: why? And isn’t these few sentences a repeat of what is said in more detail in the next section? Why is it here. Indeed why is this whole addition here, merge it with the next paragraphs.

181: “this process”: what process? Overlaying or matching?

197-199: ?? Manual mapping is simply assigning a object to a gaze point, e.g., “gaze on teapot”. What are you talking about here? You can use an abstract reference image with all your categories on there that works for all participants as it contains anything relevant to code. By the way, are these two paragraphs about Tobii Pro Lab software? That is not clear to me at all. Why discuss something you do not use (right?) in such detail?

Fig 6-8 what are true gaze coordinates and true labels? What makes them true? Just call them manual?

247: “five chunks”: frames?

247-251: what does all of this have to do with fixations at all? You’re fixing up bounding boxes of objects that are in view of the scene camera. Supposedly they are usually in view for much longer than 200 ms and this has nothing to do with the eye movement behavior. As said above, I am not sure what you are doing at all. If you are gap filling, can’t you simply find bounding box gaps not longer than a given threshold number of frames and whose size/position is not more than some threshold away and then fill that using your methods? Why arbitrarily force things into chunks?

253: what was done with 5 or more blurred images in a row? Bounding boxes discarded?

253: “the gaze was determined” what does that mean??

258: where did these images come from, taken with another camera or screenshots from the scene video?

287: 10978 frames at 25 Hz is a little over 7 minutes, not 6

293: “focusing”: looking? For the whole paragraph and other places, what are gaze focuses?

297: “this time series data could be used for analysing the study pattern and estimating the knowledge differences between participants” can it? Big claim that you do not back up, nor study.

301 and tables 1-2: what is micro and macro?

302-303: Faster R-CNN numbers don’t match table

303-304: interpolation: how do I see that in the results, that it increased accuracy?

304-307: why highlight this specific object?

312: 86%: is percent the unit for an F1 score? Also, where does this number come from?

Fig 9 is useless. Gaze shifts occur multiple times per second, but the time scale of this figure is 7+ minutes, you can see nothing meaningful at that level. The figure also is not laid out in a way that makes it easy to see similarity or differences, this is a real “find the differences” that humans are notoriously bad at…

334-343: I do not follow your logic at all, man-hours and computer hours are not at all comparable. Who cares if a computer takes a significant amount of time, let it run overnight or if it takes multiple days don’t twiddle your thumbs but do something else and you still saved all that manual coding time.

344: “better” than what?

347-357: whole paragraph seems vacuous: “This enhanced performance might be attributed to the difference in the architecture” to what else? You discuss a bunch of architecture differences, but how do you know which of these changes are important? Perhaps some made things worse? Logic and expectation is not science, yet you are claiming these laundry list of changes made things better. How do you know?

360: “square-like” rectangular?

383: “stricter” than what?

383: “threshold”: on what?

388: “In most settings, these problems cause errors that are statistical in nature, not systematic” how do you know that?

418: “robust”: YOLO v4 performs better than the others yes, but how do you make the step to that it is robust? What criterion do you apply?

Round 2

Reviewer 2 Report

I thank the authors for a thorough job replying to my comments and improving the manuscript, i can now follow much better what they did, what they claim and how they back up those claims. Some small comments remain:

unclear methods 1: I meant the computer hardware on which the model is run. I assume there is a wide variety of GPU conforming to CUDA 10.0, where performance may difference multiple-fold. Please describe the computer on which the analysis was run (GPU, CPU, memory)

unclear methods 2: I have seen that statement, but it is unclear to me. You write " For instance, if the model predicted the gaze was on the rail, the paper and the lens within one frame, but in the ground truth the gaze was only on the arrow, a false positive would be added thrice rather than once". But what if ground truth gaze would be on the rail, one hit and two false positives? Or just the hit? There are more aspects of dealing with overlapping bounding boxes to describe here

In response to your response to my "Reviewer’s comment: 197-199": Indeed manual mapping (with abstract reference image or not) would be very time consuming. Your text states its not possible ("And this makes it
211 impossible to map the gaze on the elements of a physical experiment as in our case"), and thats not correct, nor is it the big issue with manual mapping. Indeed, there are studies out there using manual mapping with recordings done in far more dynamic situations (e.g. crowds). Write instead that manual mapping requires stepping through each fixation by hand and assigning it to an object, which is very time consuming, but currently the only option (https://link.springer.com/article/10.3758/s13428-019-01314-1, https://dl.acm.org/doi/10.1145/3204493.3204568 for discussions, and the latter has some time information). I agree a better solution is needed, but your current logic doesn't hold.

In response to your response to my "Reviewer’s comment: 247-251": you are giving me a different reason than what you wrote in the text, and the logic in the text doesn't make sense to me (see previous comment). Just write that you have done the same as before and provide an appropriate reference?

In response to your response to my "Reviewer’s comment: 301": could you add this info to the manuscript as well? Ideally your work will also be read by eye tracking researchers, who are not familiar with this terminology unless they happen to also do machine learning.

Round 3

Reviewer 2 Report

I thank the authors for resolving the last issues i had (and sorry for missing the micro/macro addition).

This manuscript is a resubmission of an earlier submission. The following is a list of the peer review reports and author responses from that submission.

Round 1

Reviewer 1 Report

The paper is clearly presented and describe an interesting topic

Reviewer 2 Report

I have a big problem with the real contribution of the paper. Basically, it just shows that YOLO v4 is better than YOLO v3 for some dataset of images. It is neither original nor interesting. The dataset they used is taken by the TOBII eye tracker but it is just a camera, so we still just test a common object detection algorithm.

The authors show that object detection algorithms may be used for images registered by the eye tracker – what is obvious. They use these algorithms for some dataset of images and show that YOLO v4 is the best in detecting objects.

The only original part of the paper is the mapping of gaze points to objects. As the authors correctly suggest there are many potential problems here, even if object detection algorithms work (like overlapping bounding boxes, gaze points near the edges etc.). In my opinion extending this part of analyses (which is for now only marginally mentioned) could make the paper more interesting.

All the descriptions of algorithms (both YOLO and Faster-CNN) are vague and instead of explaining the principles concentrate on the list of elements (like SPP, PAN, RPN and so on). A person that doesn’t know these algorithms will not learn how they work and the person that already knows them will not learn anything useful. I suggest to shorten this part and concentrate on information how to train these algorithms for the specific needs.

Piece of advice: if you want to correctly find the boundaries of objects consider using Mask RCNN

Reviewer 3 Report

Mobile Eye-Tracking Data Analysis using Object Detection via YOLO v4   This paper addresses the match of detected objects in dynamic scenes by three commonly used algorithms. The work is placed in the context of a mobile eye-tracking study. That is, bounding boxes obtained by the computer vision algorithms are compared not for geometric overlap (which would be possible and a nice complement to the paper) but sampled by eye-tracking data. This is interesting, as it determines the match of the bounding boxes in a relevant application scenario.   Overall, the paper should be made better accessible for the non-expert reader. In several places, the authors jump and do not explain what might be obvious to them, but not to the reader (including me). Some examples are listed below.   Further, as the application aspect is in the foreground, the amount of work necessary for the whole setup including manual labeling of the specific application context should be quantified.   It is hard to do justice to the rapidly evolving field of computer vision / object detection in the available space. But the reference to some current overview articles would be good.   Minor comments are listed below.        
  • abstract: Here is a gap in the explanation, jumping from eye-tracking to object detection. For the non-experts mentioning Faster and Yolo does not really help. Similarly, the introduction does not really spell out the problem: In a real-world setting, how to translate the eye-tracking data to objects fixated. (Or isn't this the question you address?)
  • line 28: "needs to be done for each participant separately " Again, for the non-expert, it is not obvious why. From my point of view, in classical laboratory setups, the position of the participant relative to the scene is standardized, when you allow movements in real-world setup participants have idiosyncratic views of the world. It would be good to spend a sentence or two on the origin of the problem.
  • line 38-45. This appears not to be an introduction, but the method section.
  • line 95 and following: I fully agree with the assessment, that mapping gaze data on real-world scenes is difficult. Therefore we put our focus on virtual reality, where bounding boxes (colliders) are built into the world and allow easy identification of objects fixated (Clay et al. 2019 Eye tracking in virtual reality. J Eye Mov Res).
  • same section: the description of previous work is good. But it is not that clear what the open problem is. On which aspect does the present paper want to improve?
  • line 144: "The quality of the calibration was verified with 5 points distributed on an A3 paper. If the calibration failed, it was repeated until a successful calibration was achieved. " - What constitutes a successful calibration?
  • Figure 4b: You treat different parts of a text on one sheet of paper as different objects? This needs some explanation and discussion.
  • line 155: " In the case of manual mapping, the assignment is made by a human (person) going through all
  • 156  the fixations and indicating the gaze location on the reference image. " I do not understand this. When you have the fixation you know where the gaze is located. Or do you mean to identify the object that is fixated? You have to explain the whole reference image stuff.
  • lines 202 and following: The networks are pretrained on common databases and then trained further for the specific properties of the real-world context. This is an important constraint and should be discussed. For example, what is the amount of work for manual labeling the training videos (line 226) in comparison to manually label the experimental data?
  • equation 1: This equation is not really helpful. It is about the location of a pixel p. But P does not occur in the equation and none of the symbols is explained. Sure, I might guess. But I might guess what an optical flow algorithm is doing in the first place.
  • line 233: "We chose five chunks based on the assumption that a gaze fixation on one spot lasts at least 200 ms, which corresponds to five consecutive frames. " But the fixation might not be aligned with the boundaries of the chunks?